# Current Advances in the Bacterial Toolbox for the Biotechnological Production of Monoterpene-Based Aroma Compounds

**DOI:** 10.3390/molecules26010091

**Published:** 2020-12-28

**Authors:** Pedro Soares-Castro, Filipa Soares, Pedro M. Santos

**Affiliations:** CBMA—Centre of Molecular and Environmental Biology, University of Minho, Campus de Gualtar, 4710-057 Braga, Portugal; pcastro@bio.uminho.pt (P.S.-C.); id8342@alunos.uminho.pt (F.S.)

**Keywords:** plant volatiles, monoterpene biotransformation, valorization of plant material, microbial cell factories, essential oils, β-pinene, limonene, β-myrcene

## Abstract

Monoterpenes are plant secondary metabolites, widely used in industrial processes as precursors of important aroma compounds, such as vanillin and (−)-menthol. However, the physicochemical properties of monoterpenes make difficult their conventional conversion into value-added aromas. Biocatalysis, either by using whole cells or enzymes, may overcome such drawbacks in terms of purity of the final product, ecological and economic constraints of the current catalysis processes or extraction from plant material. In particular, the ability of oxidative enzymes (e.g., oxygenases) to modify the monoterpene backbone, with high regio- and stereo-selectivity, is attractive for the production of “natural” aromas for the flavor and fragrances industries. We review the research efforts carried out in the molecular analysis of bacterial monoterpene catabolic pathways and biochemical characterization of the respective key oxidative enzymes, with particular focus on the most relevant precursors, β-pinene, limonene and β-myrcene. The presented overview of the current state of art demonstrates that the specialized enzymatic repertoires of monoterpene-catabolizing bacteria are expanding the toolbox towards the tailored and sustainable biotechnological production of values-added aroma compounds (e.g., isonovalal, α-terpineol, and carvone isomers) whose implementation must be supported by the current advances in systems biology and metabolic engineering approaches.

## 1. The Importance of Aroma Compounds in Industry

The utilization of flavor and fragrance compounds (hereby designated as aroma compounds) is very well established in our current society, as such compounds are extensively used in the food industry (e.g., beverages, processed food, ready-to-eat meals), agrochemicals, household products (e.g., detergents, soaps), cosmetics (e.g., fine fragrances, toiletries and body care commodities) and pharmaceuticals (e.g., dietary supplements and nutraceuticals) [1]. The worldwide market for flavor and fragrance compounds was estimated to reach a value of USD 28.2 billion in 2017 and is forecast to grow at a compound annual growth rate of 5.3% up to 2023, driven by the increased demand for aroma compounds [2].

Aroma compounds are usually present in nature, mainly in plant organs, such as leaves, flowers and fruits. Their isolation involves several extraction, fractionation and purification processes (e.g., solvent extraction, distillation and chromatography) in order to obtain an extract with a high degree of purity and low chemical complexity [2,3]. Due to the limitations encountered in the current extraction methodologies from plant sources (e.g., not enough yield to satisfy the commercial demand, production costs, seasonal availability of plant raw material, risk of plant disease, variable composition of the plant extracts due to biotic and abiotic factors and stability of the extracted compounds), they have been preferentially obtained by chemical catalysis. Thus, from the market perspective, the latter synthetic compounds are most commonly used, since they are usually less expensive to manufacture and have been thoroughly tested regarding their biosafety [4,5].

In order to improve the utilization of raw materials and the reduction of chemical wastes in the manufacture and application of chemical products, the US Environmental Protection Agency coined the term “Green Chemistry” in 1990 [6]. The initiative endorsed efforts to improve or replace the conventional methodologies used in chemical synthesis, according to new guidelines which envisaged a sustainable catalysis, such as (i) the preferential utilization of renewable starting materials; (ii) the utilization of catalysts rather than stoichiometric reagents, which in turn, increase the efficiency of the synthesis; (iii) the utilization of nonhazardous and less toxic solvents; (iv) the design of shorter and more energetically efficient synthesis protocols, with milder reaction conditions; and (v) the strategical planning of the resulting reaction products and the procedures for their degradation (reviewed in [6] and references therein).

In 1997, nearly 80% of the flavors and fragrances used world-wide were produced by chemical synthesis, by multistep reactions in which the reagents were physically and chemically manipulated in order to obtain the desired compound. In the last century, great effort has been invested in the development of biological alternatives of aroma compounds, and the emergence of new chromatographic technologies allowed the separation and structural characterization of new natural molecules [7].

The trend towards the consumption of products labelled as “natural” or containing “natural compounds” has been increasing, mainly driven by the growing concern to reduce the ecological impact of the chemical industries and dependence on petroleum-based starting materials and by the need to expand the chemical catalogue of the synthetic approach [4,8]. This socioeconomic framework paved the way for the development of new strategies for the production of fine chemicals, such as top-selling flavors and fragrances [9], prompting the biotechnological exploitation of the microbial versatility to establish a sustainable bio-based economy [4,8].

Additionally, the flavoring and fragrance regulations in Europe and USA have been reviewed and adapted to encompass the scientific and technological advances in the biotechnological production of aroma compounds. According to the European and the US Code of Federal Regulations (e.g., EC 1334/2008, EC 872/2012, EC 1223/2009, US CFR 1990), compounds obtained by physical, enzymatic or microbiological processes, involving precursors isolated from vegetable, animal or microbiological origin, may be classified as “natural” [10,11].

The family of organic compounds applied in flavoring and fragrance industries includes alcohols, aldehydes, carboxylic acids, furans, fatty acids, esters, ethers, hydrocarbons, ketones, lactones, pyrazines and terpenes, with the latter representing the largest category (Table 1) [5,12,13].

Currently, several companies are investing in microbial biotechnology to establish the production of commercial important “natural” aromas using microbial biotechnology, and, in some cases, these aroma compounds are already commercialized (Table 1) [7,14,15,16,17,18,19,20]. The peach-flavored γ-decalactone was one of the first commercial aroma compounds obtained by biotechnological processes, through the bioconversion of the triglyceride 12-hydroxy-9-octadecene acid from castor seed (*Ricinus communis* L.) oil by *Yarrowia lipolytica* [20].

The establishment of the biotechnological production of vanillin is another successful example of the increasing competitiveness of “natural” aroma. Vanillin is responsible for the vanilla aroma, being one of the most popular aroma compounds (Table 1), with an estimated annual demand of approximately 20,000 tons in 2014 [15]. The commercial vanilla has been mainly produced by chemical catalysis from petroleum-derived phenol (converted into guaiacol) and glyoxylic acid, since approximately 40,000 pollinated flowers are required to collect around 500 kg of vanilla pods and extract 1 kg of vanillin [15,21]. In recent years, several companies produced vanillin from a broad range of precursors (e.g., lignin, monoterpenes eugenol and isoeugenol from clove oil, phenolic stilbenes and ferulic acid, which is a by-product of the production of rice bran oil) by patented biotechnological processes mediated by microorganisms (e.g., bacterial strains of *Escherichia coli*, *Streptomyces* spp. and *Pseudomonas putida*, as well as yeast strains of *Schizosaccharomyces pombe*) [17].

Other examples of aroma compounds whose biotechnological production is well established include nootkatone, *trans*-β-farnesene and (+)-valencene (Table 1). In particular, the utilization of microbial biotechnology has become an attractive strategy to overcome constrains of the chemical catalysis regarding very hydrophobic and volatile plant-derived aroma compounds, such as essential oils and their monoterpenic fractions.

### 1.1. Scope of the Review

The present review is intended to present an overview of the current advances in the production of monoterpene derivatives as fine chemicals, carried out by bacterial cells and bacterial enzymes. Although this work does not include and extensive description of the biological properties of the monoterpene derivatives generated in the reported bacterial metabolisms (reviewed in [1] and references therein), we highlight the main monoterpene catabolic pathways and bacterial strains currently characterized with molecular approaches as potential source of novel enzymes and novel biomolecules. Major emphasis was placed on the production of monoterpene derivatives with relevant biological activity for application in flavoring and fragrances industries, as well as for the development of agrochemicals and biocontrol agents.

The following search terms were used individually and in combination to retrieve the majority of the available and published literature on monoterpene biotechnology by bacteria: “monoterpene biotechnology”, “monoterpene biotransformation”, “monoterpene biocatalysis”, “microbial catalysis terpene”, “pinene biotechnology”, “limonene biotechnology”, “myrcene biotechnology”, “cytochrome terpene” and “cytochrome catalysis”. The information retrieved was complemented and validated by using specific search terms to target literature regarding the biotechnology of the most common monoterpene derivatives (e.g., name of compounds included in Table 1) and the monoterpene-catabolizing bacterial strains identified throughout the process. The literature surveys were performed using Google Scholar and NCBI PubMed, mainly considering original research articles, as well as review articles and the references therein. We believe that this approach provided a comprehensive overview of the field. Nevertheless, a few articles may not have been incorporated when the above search terms were not explicitly mentioned in keywords, title or abstract.

### 1.2. Aroma Compounds in Nature: Monoterpenes and Monoterpenoids

Monoterpenes belong to one of the largest classes of plant secondary metabolites—the terpenes—and are synthesized by the condensation of two molecules of C5 isoprene (2-methyl-1,3-butadiene) into a C10 hydrocarbon backbone [22,23]. Monoterpenes and their derivatives (monoterpenoids, resulting from oxygenation, methylation and esterification, among others) are volatile and highly hydrophobic compounds, predominantly found in plant essential oils.

The variability of carbon backbones (e.g., length, cyclic versus acyclic molecular structure, and the existence of functional groups) is responsible for the functional diversification of monoterpenes, which play different roles in the physiology of plants, such as in plant defense against microbial, insect and herbivores attack, as well as in plant-to-plant communication and pollinator attraction [22,23]. This functional diversity is responsible for the broad range of organoleptic and therapeutic properties observed. Monoterpenoids are usually less hydrophobic and less volatile than their monoterpene counterparts, whereas the properties of the original molecule are retained or even enhanced [7,24]. Therefore, monoterpenoids have great industrial relevance and allow a more stable incorporation in aqueous formulations. As value-added compounds, monoterpenoids usually have a higher market value (e.g., in 2017, de Oliveira Felipe et al. reported that (*R*)-(+)-limonene presented a reference price of USD 34 L^−1^ whereas the market price of the derivatives carveol, carvone and perillyl alcohol was USD 529 L^−1^, USD 350 L^−1^ and USD 405 L^−1^, respectively [4].

Monoterpenes and monoterpenoids obtained from natural sources are considered as GRAS (generally recognized as safe) compounds and can be added to products without being considered as artificial additives. This status was given in 1965, being reviewed and reaffirmed as GRAS in 2010, based on studies concerning the rapid absorption, metabolic conversion and elimination in humans and animals, as well as in studies of subchronic and chronic pathologies showing the lack of significant genotoxic and mutagenic potential, thus without posing any significant risk to human health [25].

## 2. Catalysis Mediated by Biological Systems

The biological-based production of value-added compounds, either using whole cells or purified enzymes, offers additional promising advantages when compared to the chemical catalysis. The main advantages of microbial biotechnology over chemical methods include (i) the ability to produce compounds, often with higher regio- and stereo-selectivity, especially in enzyme-mediated catalysis (e.g., selective oxygenation of C=C or C–H bonds in the hydrocarbon backbone); (ii) a higher reaction efficiency which is translated in the requirement of smaller amounts of catalyst; (iii) the utilization of mild reaction conditions of temperature and pH (usually ranging from 20 to 40 °C and from pH values of 5 to 8), which minimizes undesired side reactions such as decomposition, isomerization, racemization and rearrangements of the chemical species (reviewed in [6] and references therein).

Moreover, different enzymes usually require compatible reaction conditions which allow the setup of multienzyme systems for reaction cascades in order to simplify the catalytic process. In some contexts, the utilization of microbial cells as catalysts might present some advantages. Cells, as biofactories, provide an adequate environment and cofactors for an optimal enzymatic stability and activity, especially when a multistep catalysis is required. Additionally, different microbial cells/species can be used as communities, to perform functions which would otherwise endow heavy metabolic burden for individual cells (e.g., independent pathways competing for intracellular resources, and accumulation of intermediates to high/toxic concentrations) [26]. The advantages of the whole cell-based catalysis is also associated with the ease of microbial cultivation, speed of growth supported by the use of less expensive substrates, which, in many cases, might be wastes/by-products of other industrial processes, and the capability to be genetically altered [6,27].

The biotechnological production of monoterpenes and monoterpenoids as aroma compounds may be accomplished by single-step biocatalysis and multistep biotransformation of a precursor into a specific product or de novo synthesis from simple building block molecules as carbon sources, which are converted into more complex biomolecules through anabolic pathways [4,14,17]. At the first stage, the biological system, and particularly whole cells, must come into contact with the substrate and remain stable to catalyze the transformations in the monoterpene backbone.

### 2.1. Bacterial Adaptation to the Hydrophobicity of Monoterpene Substrates

The hydrocarbon structure of monoterpenes and monoterpenoids provides them with high hydrophobicity and, similarly to other membrane-active organic compounds like alkanes, they are capable of easily altering the membrane fluidity (usually increasing it) by accumulating within the acyl chains of phospholipids [28,29,30]. The cell membrane is a permeability barrier against harmful compounds and acts as a matrix for proteins (signaling, transport and enzymes) and energy transduction processes [31,32]. Consequently, hydrophobic organic compounds may inactivate membrane-associated proteins (e.g., ATPases and ion channels/pumps essential for energy and ion homeostasis), cause leakage of ions and intracellular macromolecules due to changes in the permeability and integrity, abolish membrane potential and even alter the proton gradient, which results in an altered intracellular pH [28]. Ultimately, the loss of membrane function leads to impairment of cell metabolism, inhibition of growth and cell death.

The toxicity derived from hydrophobic organic compounds is correlated with the logarithm of its octanol/water partition coefficient (LogP or Log K_ow_), in which LogP values ranging from 1 to 5 usually translate into compound toxicity for whole cells [28,31]. As an adaptive response, cells may change the membrane fatty acid profile to preserve membrane characteristics. Nevertheless, the mechanisms reported seem to be dependent on the genomic background of the cell and cell physiology (e.g., the growth phase), but also on the chemical properties (e.g., acyclic vs. cyclic, chain length, branching degree, and the degree and position of the oxyfunctionalization) and concentration of the stressor ([30,33,34,35] and references therein).

In general, facing an instant increase in the membrane fluidity and swelling due to the partitioning and accumulation of organic compounds into the lipid bilayer, cells might behave in the following ways:I.They may increase the degree of saturation of fatty acids by de novo synthesis of new molecules when cell growth is not inhibited. The increase in saturated acyl chains leads to a denser membrane packing, thereby improving the tolerance towards organic compounds [31,32].II.They may alter the length of the acyl chains by increasing the synthesis of fatty acids with longer acyl chains. Longer lipid molecules have higher melting temperatures; therefore, the conversion of the membrane from a lamellar gel to liquid crystalline phase is hindered [31,32].III.They may alter the profile of phospholipid head groups, which is predicted to influence the physical and chemical properties of the membrane (e.g., charge and melting point) [31].IV.They may swiftly increase cell surface hydrophobicity by altering the composition of the lipopolysaccharide layer (e.g., complete loss of B band lipopolysaccharide), and generate outer membrane vesicles (OMVs). Although this ubiquitous mechanism has not been extensively reported as a response to hydrophobic stress, several studies with solvent-tolerant *P. putida* strains have shown the induction of vesiculation mediated by alkanes and alkanols ([36] and references therein). This strategy may provide an enhanced ability for protective cell attachment, aggregation and biofilm formation, as well as for partitioning the hydrocarbon stressor in vesicles ([37] and references therein).V.Several bacterial strains (e.g., *Pseudomonas* spp., *Vibrio* spp., strains of *Methylococcus capsulatus*, *Alcanivorax borkumensis* and *Colwellia psychrerythraea*) employ a fifth adaptive mechanism by isomerizing *cis*-unsaturated fatty acids to *trans*-unsaturated acyl chains ([37] and references therein). *Cis*-unsaturated acyl chains comprise a bend of 30°, which disturbs the ordered fatty acid packing, increasing fluidity, allowing denser packing and promoting an increase in membrane stiffness to counteract excessive fluidity ([37] and references therein). This membrane *cis*-to-*trans* isomerization is performed by *cis/trans* isomerases, and since it is dependent on neither energy nor on the de novo synthesis of fatty acid molecules, this mechanism is considered a rapid short-term response to chemical stress.

### 2.2. Mechanisms for the Bacterial Transformation of the Hydrocarbon Backbone

In bacteria, the biotransformation of monoterpenes and monoterpenoids is carried out by similar mechanisms described for the catabolism of the medium-chain hydrocarbon backbone (C5 to C12 chain) of *n*-alkenes and *n*-alkanes [38], which involves multiple oxidative steps with the subsequent formation of alcohols, aldehydes/ketones and fatty acid-like molecules activated with a coenzyme A moiety (CoA). The CoA derivatives are then channeled into the central metabolism by a β-oxidation-like enzymatic machinery, yielding a C2 acetyl-CoA molecule and a fatty acid with two less carbon atoms than the original substrate [39] (Figure 1).

Although the β-oxidation mechanism is common among bacteria, the strategies adopted by hydrocarbon-catabolizing bacteria to initiate the catabolism of this type of carbon sources greatly differ among strains (Table 2), according to their genomic background and physiology, the acyl chain length, degree of saturation and backbone branching or cyclization (reviewed in [40,41] and references therein).

#### 2.2.1. Molecular Mechanism for the Catabolism of the Unsaturated Hydrocarbon Backbone

The catabolism of acyclic or cyclic medium-chain unsaturated hydrocarbons starts either with the formation of a hydroxyl group into a C–OH bound or by the epoxidation of a double bond [41] (Figure 2). The first oxygenation of the nonpolar backbone is usually the rate-limiting step in the catabolism of hydrocarbons.

Particularly, the aerobic catabolism of acyclic hydrocarbons might be initiated by (i) terminal hydroxylation [41,48,49], (ii) hydroxylation at both termini (ω-oxidation) which results in dioic backbones [41,50], (iii) sub-terminal oxidation into secondary alcohols (α-oxidation) [40] or (iv) Finnerty oxidation mediated by a dioxygenase [51]. Moreover, the epoxidation of an unsatured bound generates an epoxide ring, which might react with cellular biomolecules (e.g., DNA and proteins; Figure 2). Therefore, the epoxide ring may be further cleaved into an alcohol by several mechanisms that neutralizes its reactivity, namely (i) by the activity of epoxide hydrolases (e.g., conversion of 3,4-epoxybutyrate by an *Acinetobacter baumannii* strain [52] or conversion of limonene-1,2-epoxide to limonene-1,2-diol by the *Rhodococcus erythropolis* DCL14 strain [53]); (ii) by the conjugation with the thiol group of glutathione, mediated by the redox scavenging system of glutathione transferases [54] (e.g., catabolism of isoprene monoxide and aliphatic epoxides by *Rhodococcus* sp.); or (iii) by the carboxylation of the epoxide bound with carbon dioxide to form a keto acid (e.g., conversion of 2,3-epoxybutane by an epoxide carboxylase of the *Xanthobacter* sp. Py2 strain [55]).

The catabolism of cyclic hydrocarbons in aerobic bacteria, such as the monoterpene-related cyclohexane backbone, is initiated by sequential oxidations to form a cyclic ketone (Figure 2). The following catabolic step is mediated by a Baeyer–Villiger monooxygenase, which catalyzes the oxygen insertion into the cyclic ring and generates a caprolactone [56]. The ring is cleaved by a lactone hydrolase into an acyclic oxygenated compound, which can be degraded by enzymatic systems acting on acyclic hydrocarbons [41].

The anaerobic degradation of unsaturated hydrocarbons mainly involves the hydration of the double bond to a saturated alcohol to allow catalysis by the *n*-alkane-degrading enzymes (Figure 2) [57,58]. In anaerobic bacteria, two main mechanisms have been reported for the catabolism of acyclic *n*-alkanes, which include the addition of a fumarate moiety (usually at the sub-terminal C2 atom) or a carboxyl moiety derived from bicarbonate to the *n*-alkane backbone [57,59]. The catabolism of the cyclohexane backbone in the absence of molecular oxygen has been proposed to include the generation of a ketone, whose ring is cleaved by hydrolysis of the C–C bond [60].

#### 2.2.2. Molecular Mechanism for the Catabolism of the Branched Hydrocarbon Backbone

As a result of isoprene condensation, monoterpenes have branched backbones. The substitution of protons near the termini of an alkyl hydrocarbon chain, particularly in the β-carbon (3-methyl branched) or a quaternary branch, is described to sterically hamper the activity of the enzymes of the β-oxidation pathway [41,61]. Therefore, alternative strategies are used by monoterpene-catabolizing microorganisms to mineralize the branched backbone (Figure 2). The 2-methyl branched hydrocarbons have been reported to undergo terminal oxidation (e.g., ω-oxidation) similarly to *n*-alkane substrates, yielding mono- or di-carboxylic acids. The β-oxidation-like mechanism of this branched terminal is associated with the release of a propionyl-CoA moiety, rather than the common acyl-CoA molecule, which might be channeled to the central metabolism through the 2-methyl citrate cycle [62]. In the catabolism of 3-methyl branched hydrocarbons, the carbon source backbone might undergo a β-decarboxylation to allow the degradation through the β-oxidation pathway (Figure 2). The catabolism of geranic acid (3,7-dimethylocta-2,6-dienoic acid) by the acyclic terpene utilization pathway has been the most well-characterized example of this mechanism [62,63,64]. The geranic acid is converted to geranyl-CoA, and its β-methyl group is converted to an acetate group by carboxylation, yielding isohexenyl-glutaconyl-CoA [65]. After addition of a hydroxyl group to resolve the double bond in the C3 position of isohexenyl-glutaconyl-CoA, an acetate moiety is released, and the β-oxidation of the resulting 7-methyl-3-oxo-6-octenoyl-CoA molecule is resumed.

The anaerobic catabolism of 2-methyl branched hydrocarbons is proposed to occur via similar mechanisms of the fumarate addition pathway for saturated backbones, followed by decarboxylation and activation with a CoA moiety [66].

The monoterpene-catabolizing strains have evolved towards the mineralization of this type of carbon source, and, thus, they may be considered promising biological systems for monoterpene biotransformation, harboring specialized chemosensory and enzymatic machinery for the oxidative catalysis of the monoterpene backbone, as well as being capable of employing suitable physiological mechanisms to cope with and adapt to the high hydrophobicity of these hydrocarbons.

### 2.3. Nature’s Reservoir of Bacterial Biocatalysts for Industrially Relevant Monoterpenes

This section reviews the reports of monoterpene catabolic pathways in bacteria (Appendix A), with particular focus on the biotransformation the main monoterpene precursors: the bicyclic α- and β-pinene, the cyclic limonene and the acyclic β-myrcene.

#### 2.3.1. Pinene Isomers: A Bicyclic Precursor

The pinene isomers (α- and β-pinene) are the most important monoterpenes in industry. The α- and β-pinene represent 75% to 90% of the essential oil from conifers and can be found in concentrations ranging from 50% to 70% and from 15% to 30%, respectively, in turpentine oil (crude resin from conifer trees, especially pine trees), a by-product of the paper and cellulose industry produced abundantly (e.g., worldwide production estimated to reach 330,000 tons per year in 2013) [24,67]. Thus, α- and β-pinene are an inexpensive and abundant monoterpene material. These compounds are the most abundant bicyclic monoterpenes and can be precursors for derivation into virtually all monoterpenes used in industrial applications, namely aroma compounds widely used in food and cosmetic industries. The main derivatives directly obtained by chemical catalysis comprise borneol, camphor, camphene, limonene, terpineols and terpinolene, carvone, verbenol, verbenone and the acyclic β-myrcene (Figure 3) [68,69], which in turn can also be used as precursors for the production of a broader array of molecules (see the following Section 2.3.2 and Section 2.3.3 regarding limonene and β-myrcene, respectively).

Different pathways have been reported in the literature for the bacterial degradation of pinene isomers (Figure 4).

The epoxidation of α-pinene into α-pinene oxide was the first pathway proposed in bacteria, mainly studied in *Pseudomonas rhodesiae* CIP 107491 and *Sphingobium* sp. NCIMB 11671 (former *Pseudomonas fluorescens* NCIMB 11671) strains. The ring cleavage of α-pinene oxide, mediated by a putative α-pinene oxide lyase (the Prα-POL in CIP 107491 strain), resulted in the formation of the aroma compounds isonovalal and novalal (Figure 4b,c) as major biotransformation products in *P. rhodesiae* CIP 107491 and *Sphingobium* sp. NCIMB 11671, respectively [74,80]. The optimization of α-pinene oxide biotransformation in a *n*-hexadecane two-phase system, with sequential feedings of permeabilized CIP 107491 cells and precursors, led to a maximum recover of 400 g L^−1^ of isonovalal [81]. Higher titers of 540 g L^−1^ of isonovalal were further achieved with the constitutive heterologous expression of Prα-POL in *E. coli* BL21(λDE3) in similar fed-batch biotransformation conditions, thereby overcoming latency phases of preliminary induction and endogenous regulatory pathways in *P. rhodesiae* CIP 107491 [82]. A similar mechanism of ring cleavage was also proposed in the *Nocardia* sp. P18.3 strain, although the epoxide-forming α-pinene monooxygenase was never identified in any of the three strains [75].

Savithiry et al. (1998) [70] reported the catabolism of α- and β-pinene by the thermophilic *Bacillus pallidus* BR425 strain, with the production of bicyclic and monocyclic intermediates, via pinocarveol/myrtenol or limonene (Figure 4d–h). β-Pinene, pinocarveol, pinocarvone, myrtenol, myrtenal, limonene, and carveol were detected during the growth of BR425 cells with an organic phase of α-pinene, which suggested a link between different pathways and the ability of *B. pallidus* BR425 enzymatic repertoire to oxidized pinenes at different positions [70] and generate diverse compounds with a broad application as flavors, components of fragrances and of pharmaceuticals.

Industrially attractive monocyclic intermediates were also detected in biotransformation studies (Figure 4d–h) with the *Pseudomonas* sp. NCIMB 10684 strain (e.g., *trans*-carveol) [73], a *Serratia marcescens* strain (e.g., *trans*-verbenol, verbenone, *trans*-sobrerol, and α-terpineol) [79], the *Pseudomonas* sp. NCIMB 10687 strain (e.g., *p*-cymene, limonene, α-terpinolene, α-terpineol, and borneol from both α- and β-pinene isomers) [76] and *Pseudomonas veronii* ZW (e.g., *p*-cymene, limonene and myrtenol from α-pinene) [77].

#### 2.3.2. Limonene: The Monocyclic Precursor

Limonene exists in nature as two enantiomers, (*R*)-(+)- and (*S*)-(−)-limonene, the former being the most abundant isomeric form in plants. Although limonene can be synthesized from pinene isomers, it is present in high concentrations in orange peel oil (approximately 90%) and can be obtained in large amounts as by-product in the production of citrus juice and pulps (e.g., worldwide production estimated to reach 30,000 tons per year in 2013) [24,83,84]. The monocyclic structure of limonene is an inexpensive precursor for the production of oxygenated derivatives, also with a broad range of applications due to their odorant and bioactive properties: carveol, carvone isomers, perillyl alcohol, menthol, *p*-cymene and α-terpineol, among others (Figure 3) [83,84].

In bacteria, there are four main alternative pathways described for the catabolism of limonene (Figure 5). The most well-described model for limonene catabolism is the *R. erythropolis* DCL14 strain (Figure 5a,b). *R. erythropolis* DCL14 is able to oxidize both limonene enantiomers at the 1,2-double bond via the activity of a limonene-1,2-monooxygenase LimB and limonene-1,2-epoxide hydrolase LimA, which results in limonene-1,2-epoxide and limonene-1,2-diol, respectively, with 100% conversion yield [85,86]. The aroma limonene-1,2-diol is hypothesized to be subsequently oxidized into the ketone 1-hydroxy-2-oxolimonene by a dehydrogenase and converted to a lactone by a Baeyer–Villiger monooxygenase (1-hydroxy-2-oxolimonene 1,2-monooxygenase) for further ring cleavage, activation with coenzyme A moiety and degradation by the β-oxidation-like pathway. The generation of lactones is frequent in the bacterial catabolism of bicyclic and monocyclic monoterpenes, catalyzed by Baeyer–Villiger monooxygenases (BVMOs). These cyclic monoesters are present in all major classes of foods (e.g., fruits and vegetables, nuts, meat, milk products and baked products), in which they may contribute to flavor nuances [87].

The strain DCL14 can also degrade (*R*)- and (*S*)-limonene through a second pathway, in which the substrate is hydroxylated in the 6-position by a putative limonene-6-monooxygenase and generates (*R*)- and (*S*)-carveol. DCL14 cells can oxidize both (*R*)- and (*S*)-stereoisomers to carvone and dihydrocarvone, mediated by the carveol dehydrogenase LimC and a putative carvone reductase, respectively [97]. Although the conversion of carveol isomers to flavoring carvone isomers has previously been detected in other Gram-positive strains [70,98], the putative enzymes catalyzing the reaction have never been characterized. In *R. erythropolis* DCL14, the monocyclic structure of (iso-)dihydrocarvone isomers, may then be cleaved through a similar mechanisms acting on 1-hydroxy-2-oxolimonene: oxygenation by a BVMO into a lactone and subsequent lactone hydrolysis to generate 6-hydroxy-3-isopropenylheptanoate or 6-hydroxy-5-isopropenyl-2-methylhexanoate from (1*R*,4*R*)- or (1*S*,4*R*)-(iso-)dihydrocarvone, respectively [97]. Notably, this ring-opening step is mediated by the monoterpene ɛ-lactone hydrolase MlhB, which displays a broad substrate range for lactones derived not only from the biotransformation of carvone, but also from isopulegol and menthol/menthone, among others [99]. The extent of limonene biotransformation by DCL14 cells via limonene-6-monooxygenase (the carveol/carvone-forming pathway) was reported to be dependent on the cytotoxicity threshold of the accumulated carveol and carvone. Morrish and co-workers [100] were able to enhance the product titers in a fed-batch two-phase partitioning bioreactor with silicone oil, which allowed them to achieve up to 0.32 g L^−1^ of carvone by increasing the amount of (+)-limonene used as C-source and of (−)-carveol used as substrate for the enzymatic conversion.

Limonene biotransformation can also occur via hydroxylation of the primary methyl group (in the C7 position) into the aroma perillyl alcohol [101] (Figure 5c), whose availability in nature is limited. Given the anticarcinogenic property of perillyl alcohol and its derivative perillic acid, this pathway has been studied in members of several bacterial genera, mainly aiming at the characterization of promising biocatalysts. The pathway was initially described in *Geobacillus stearothermophilus* BR388 (former member of *Bacillus* genus) [90], and the following research studies were focused on the optimization of the hydroxylation with heterologous cytochrome CYP153A6 system from *Mycobacterium* sp. HXN-1500 in *P. putida* and *E. coli* strains, as well as in *Castellaniella defragrans* DSM 12143 [89,92,102,103]. The genome sequencing, protein and metabolite profiling coupled to mass spectrometry, as well as transposon mutagenesis approaches, located the key genes of the anaerobic metabolism of cyclic monoterpenes from *C. defragrans* DSM 12143 in a 70 kb *locus*, including four genes coding for a putative limonene dehydrogenase system: the dehydrogenase subunits CtmA and CtmB, the ferredoxin CmtE and ferredoxin reductase CmtF [89]. The highest titer of perillic acid was reported in *P. putida* DSM 12264 [92,104], whose cultures produced up to 31 g L^−1^ after 7 days in a fed-batch bioreactor, by coupling the biotransformation process with an in situ product recovery step based on anion exchange resin to overcome perillic acid-mediated cellular inhibition.

Bacterial enzymatic machinery can also hydrate the limonene isoprenyl double bond (in the C8 position) into the versatile α-terpineol [105] (Figure 5f). The highest titers of biotechnologically produced α-terpineol have been reported in biotransformation experiments with the bacteria *Sphingobium* sp. NCIMB 11671 by using two-phase bioreactor systems. After optimizing pH, biocatalyst concentration, substrate concentration, the aqueous:organic ratio, pH, temperature and agitation, approximately 240 g of (*R*)-(+)-α-terpineol was produced per liter of sunflower oil organic phase from (*R*)-(+)-limonene, with a 94.5% transformation yield [95,106].

Although α-terpineol seemed to be a dead-end product in *Sphingobium* sp. NCIMB 11671 and other limonene-transforming strains, the hydroxylation of α-terpineol to 7-hydroxyterpineol was observed in an unclassified strain of the *Pseudomonas* genera, catalyzed by the cytochrome CYP108A1 system (P450_terp_) [107,108].

In some bacterial strains, products not corresponding to the previous described pathways were reported: Production of the flavor compounds γ-valerolactone and cryptone, linalool and dihydrolinalool, (Figure 5d,g) by the *Kosakonia cowanii* 6L strain (former member of the *Enterobacter* genus) [91]; the epoxidation of the 8,9-double bond of limonene into limonene-8,9-epoxide, a volatile found in mandarin and ginger, by the cyclohexane-grown *Xanthobacter* sp. C20 strain (Figure 5e) [94]; and the production of sobrerol by the *P. putida* MTCC 1072 strain [93], among others (Figure 5g).

#### 2.3.3. β-Myrcene: The Versatile Acyclic Precursor

Currently, β-myrcene is mainly obtained from the pyrolysis of β-pinene [68]. The acyclic and unsaturated structure of β-myrcene (e.g., 1,3-diene moiety) can undergo a number of reactions, such as isomerization and cyclization, and, therefore, it is used as a precursor for the production of many monoterpene derivatives [68], cyclic or acyclic, unsaturated or saturated, and with diverse functional groups (Figure 3).

An overview of the main reported pathways for the bacterial catabolism of β-myrcene and other acyclic monoterpenoids is presented in Figure 6.

The first studies focused on the bacterial catabolism of β-myrcene involved the *P. putida* S4-2 [109] and *Pseudomonas* sp. M1 strains [110]. Cultures of *P. putida* S4-2 initiated the mineralization of β-myrcene by hydroxylation of the terminal C8-position, yielding (*E*)-myrcen-8-ol (Figure 6a), with subsequent production of the (*E*)-2-methyl-6-methylene-2,7-octadienoic acid (myrcenoic acid), 4-methylene-5-hexenoic acid and (*E*)-4-methyl-3-hexenoic acid [109].

Iurescia and co-workers [110] reported the first molecular insights regarding the genetic code underlying the β-myrcene catabolism via terminal hydroxylation as a result of the isolation of a *Pseudomonas* sp. M1 mutant (strain M1–N22) generated by *Tn*5 random mutagenesis. In strain M1–N22, the *Tn*5 transposon interrupted the alcohol dehydrogenase-coding gene *myrB*, and impaired growth using β-myrcene as sole carbon and energy sources, leading to the accumulation of (*E*)-myrcen-8-ol as the major β-myrcene derivative during biotransformation experiments. The *myr + wt* phenotype of strain M1–N22 was restored by genetic complementation in a screening carried out with a M1-derived genomic library, which led to the identification of four genes, organized as the cluster *myrDABC*, potentially coding for β-myrcene-biotransforming enzymes. Based on the work of Iurescia et al. and of Narushima et al. [109,110], the first catabolic pathway present in both *Pseudomonas* strains was proposed to be initiated by the hydroxylation of β-myrcene in the terminal carbon into myrcen-8-ol, followed by the oxidation to aldehyde by the alcohol dehydrogenase MyrB, subsequent oxidation to the carboxylic acid by the putative aldehyde dehydrogenase MyrA and activation by a coenzyme A moiety, involving the putatives acyl-CoA synthetase MyrC and enoyl-CoA hydratase MyrD, which would allow the channeling of β-myrcene intermediates to the central metabolism by an enzymatic cascade analogous to the beta-oxidation pathway.

In the last decade, *Pseudomonas* sp. M1 has been characterized using a systems biology approach at the genome, transcriptome and proteome levels in the scope of its outstanding ability to mineralize β-myrcene [120,121]. The characterization of the β-myrcene stimulon by RNA-seq identified a novel 28 kb genomic island (GI), not previously reported in any other biological system, whose expression was strongly stimulated in the presence of β-myrcene [121]. Consequently, the molecular characterization of the gene products coded by the 28 kb GI suggested their putative involvement in the following: (i) monoterpene sensing; (ii) regulation of gene expression; and (iii) β-myrcene oxidation and bioconversion of β-myrcene derivatives into central metabolism intermediates [121], including the previously identified *myrDABC* cluster [110].

Soares-Castro et al. carried out a metabolic footprint analysis of cultures of strain M1 *wt* supplemented with β-myrcene over time, and they detected (*E*)-myrcen-8-ol, myrcene aldehyde and the myrcenoic acid at 30 min of β-myrcene biotransformation [30], thus supporting the previously proposed β-myrcene catabolic pathway. Two putative derivatives of the myrcenoic acid were detected in strain M1 *wt* supernatants after 3.5 h of β-myrcene supplementation: (*E*)-4-methyl-3-hexenoic acid and the 4-methylhexanoic acid [30]. The former was suggested by Narushima et al. [109] as a product of a β-oxidation-like catabolic step targeting the myrcenoic acid with subsequent elimination of the carboxyl-containing C3-unit as propionyl-CoA, whereas the latter might result from the dehydrogenation of the C3,C4 double bond of (*E*)-4-methyl-3-hexenoic acid. From the set of β-myrcene derivatives produced by strain M1, myrcen-8-ol and 4-methylhexanoic acid have been reported to be insect pheromones and may be exploited as biocontrol agents [122,123], whereas 4-methyl-3-hexenoic acid is a described fragrance compound [124].

Moreover, the molecular and metabolic characterization of the *myr*- strains M1–C19 and M1–C38, obtained by *Tn*5 transposon mutagenesis to abolish the β-myrcene-inducible expression of the β-myrcene core code, led to the identification of two gene products essential to induce the 28 kb-encoded β-myrcene catabolic machinery and confer the ability to use this monoterpene as sole carbon source in M1 cells [30]: (i) the LuxR family transcriptional regulator MyrR (PM1_0322860), homologous to members of the MalT subfamily and which may act as a sensory switch-like modulator of β-myrcene-inducible gene expression; (ii) the membrane β-myrcene hydroxylase MyrH (PM1_0322855), which is involved in the conversion of β-myrcene to myrcen-8-ol. Strikingly, the expression of the GI modules also resulted in the successful biotransformation of other cyclic and acyclic terpene backbones, originating oxidized metabolites, some of which with great biotechnological potential. M1 cells biotransformed (−)-β-pinene into α-terpineol and citronellic acid, whereas the cyclic structure of (*R*)-(+)-limonene was hydroxylated into limonene-1,2-diol. Linalyl acetate and β-linalool were biotransformed into geraniol and geranic acid, suggesting a common catabolic pathway with β-citronellol via the acyclic terpene utilization pathway encoded in the M1 genome and extensively characterized in *Pseudomonas citronellolis* DSM 50332 and *Pseudomonas aeruginosa* PA01 [65,118].

Esmaeli and Hashemi (2011) described the conversion of β-myrcene in the *P. aeruginosa* PTCC 1074 strain, which produced dihydrolinalool, 2,6-dimethyloctane and the cyclic α-terpineol [111] (Figure 6b). The overlapping degradation of β-myrcene with the catabolic pathways of other acyclic alcohols (linalool, geraniol) has also been described [64,112,125] (Figure 6c,d). The anaerobe *C. defragrans* DSM 12143 codes a linalool dehydratase–isomerase [58,125], which can catalyze the reversible hydration of β-myrcene to (*S*)-(+)-linalool and isomerization of (*S*)-(+)-linalool to geraniol, apparently dependent on the equilibrium of hydrocarbon-to-alcohol concentrations. Geraniol is further oxidized to geranic acid by two identified geraniol and geranial dehydrogenases, GeoA and GeoB, respectively [58]. A similar isomerization mechanism of linalool to geraniol, followed by subsequent oxidation to geranial and geranic acid, was proposed for the anaerobe *Thauera linaloolentis* DSM 12138 [64]. Geraniol was also detected in cultures of *R. erythropolis* MLT1 supplemented with β-myrcene as the sole carbon source, although the enzyme involved in β-myrcene oxygenation was not identified [112].

The production of important aroma compounds were also reported in the catabolism of linalool, β-citronellol and geraniol by other bacterial strains, such as *P. putida* ATCC 29607, the cytochrome-rich *Novosphingobium aromaticivorans* DSM12444 and the menthol-producing *Pseudomonas convexa* (Figure 6j) [114,115,117,119].

From the biotechnological perspective, once a particular pathway for monoterpene biotransformation is identified and characterized, the next step involves the establishment of the necessary knowledge to develop a metabolic engineering approach of such pathway in a way that the biotransformation, rather than a complete mineralization, becomes optimized and tailored for specific applications [69].

## 3. Exploiting the Biotechnological Potential of Monoterpene-Catabolizing Enzymes

The majority of monoterpene derivatives with industrial relevance has been detected as metabolic intermediates in the reported catabolic pathways (presented in the Section 2), suggesting that the repertoire of enzymes available in the environment is able to mimic the current chemical synthesis reactions. A few biological systems are already used as monoterpene biocatalysts (whole cells and individual enzymes) in industrial processes, mainly focused on the production of the aroma compounds vanillin, carvone and (−)-menthol (Table 2).

The literature revision presented in Section 2 also highlighted several other functional modules that have been the focus of molecular and biochemical characterization, with potential for the development of biotechnological tools (Appendix A).

The biochemical and structural characterization of enzymes from monoterpene-catabolizing bacteria have been providing the necessary knowledge required for protein engineering. In order to achieve an efficient and economically feasible biotechnological process, protein optimization envisages the improvement of the catalytic productivity, broadening pH and temperature range for enzyme activity, enhancement of substrate selectivity, enhancement of protein solubility and long-term stability, changing regio- or stereo-specificity and changing cofactor requirements [126,127]. Enzyme engineering can also focus on altering cofactor dependence, reduce catalytic inhibition derived from accumulation of end-products and even expand the substrate range to other natural or non-natural molecules [128,129]. This optimization can be performed by evolutionary approaches; rational protein design by directed mutagenesis towards key residues or domains; de novo generation of enzymes based on computational algorithm; in silico modelling; and creating chimeric proteins or utilizing protein, DNA or RNA scaffolds [130,131,132].

The cytochrome P450_cam_ (CYP101A1) of *P. putida* ATCC 17453 and *P. putida* ATCC 29607, involved in the hydroxylation of camphor to 5-exo-hydroxycamphor (Appendix A), is one of the most extensively studied members of the cytochrome family, being an archetypal model for structural and mechanistic studies. Several engineering strategies were carried out to modulate the substrate range of P450_cam_ to non-native compounds by substitution of key amino acid residues (e.g., the mutation Y96F translates into a variant that better accommodates nonpolar monoterpenes in the active site) [133]. Some engineered variants of P450_cam_ were able to oxidize other monoterpenes (e.g., α-and β-pinene, (+)-carene and 1,8-cineole (*S*)-limonene, among others) [133,134], polycyclic aromatic hydrocarbons (e.g., naphthalene, pyrene, phenanthrene and fluoranthene) [135,136], polychlorinated benzenes [136], and even aliphatic gaseous alkanes (ethane, propane and n-butane) [134,137]. Similarly, the directed evolution of the cytochrome P450_BM-3_ (fatty acid hydroxylase CYP102A1) of *Bacillus megaterium* ATCC 14581 allowed the generation of variants with altered substrate specificity towards the hydroxylation of different monoterpenes (Appendix A), including the differential conversion of α-pinene into α-pinene oxide, the insect pheromones *trans*-verbenol and verbenone, myrtenol and sobrerol, the oxidation of (*R*)-limonene into (*R*)-limonene-8,9-epoxide (previously detected in the biotransformation with *Xanthobacter* sp. C20) or perillyl alcohol and the epoxidation of geraniol into 6,7-epoxygeraniol (a precursor of furanoids used in fragrances) [132,133,138,139].

Several other cytochromes have been characterized throughout the year, in different bacterial genera, showing that members of the class I of bacterial P450 cytochromes are often highly active and selective, making them suitable for the oxyfunctionalization of the monoterpene backbone. The engineering approaches carried out on bacterial cytochromes has greatly benefit from the increasing knowledge generated on P450 cytochrome biochemistry and crystallography, including studies on other well characterized members involved in monoterpene catabolism, such as the P450_terp_ (CYP108A1) and P450_cin_ (CYP176A1) of *Citrobacter braakii*, and the P450_lin_ (CYP111A1) of *P. putida* ATCC 29607, among others (Appendix A) [107,132,133,140]. Currently, the members of class I P450 cytochromes and their variants comprise one of the most well characterized toolbox for bacterial biotechnology.

Another example is the characterization effort that has been carried out with the limonene-1,2-epoxide hydrolase from the *R. erythropolis* DCL14 strain, which is structurally different from other members of the same class, thereby translating into an epoxide-cleaving mechanism with different stereoselectivity [141]. Besides limonene-1,2-epoxide, limonene-1,2-epoxide hydrolase was able to transform other substrates, including cyclic (e.g., 1-methylcyclohexene oxide, cyclohexene oxide and styrene-7,8-oxide) and aliphatic backbones (e.g., 2-methyl-1,2-epoxides) [141,142], albeit showing lower activity levels. Engineered variants of the cofactor-independent linalool dehydratase–isomerase Ldi of *C. defragrans* DSM 12143 have been explored for the enzymatic dehydration of short-chain alkanols into volatile dienes, such as butadiene and isoprene [143]. These studies highlight the potential versatility and substrate modulation of such enzymes, aiming at the production of unprecedented compounds and development of new catalytic processes.

Cyclic products have been detected as a result of the catabolism of some acyclic monoterpenes, such as of β-myrcene, citronellol and linalool [111,117,119,144]. The bacterial monoterpene cyclases, which mimic the cyclization reactions performed in organic chemistry and plant synthesis, also comprise an attractive class of enzymes for biocatalysis [145,146]. Although this class of enzymes appears to be annotated in several microbial genomes, very little is known about their molecular and biochemical mechanisms in bacterial, particularly in regard to using the monoterpene backbone as substrate for the production of important cyclic aroma compounds, such as menthol and carvone. Bastian et al. (2017) generated engineered variants of the squalene hopene cyclase from *Alicyclobacillus acidocaldarius* (AacSHC), which was able to catalyze the selective cyclization of (*S*)-citronellal and (*R*)-citronellal to the different stereoisomers of isopulegol, including (−)-iso-isopulegol, which is a key precursor for the production of the aroma compound (−)-menthol [146].

### 3.1. Metagenomics Approaches May Expand the Bacterial Toolbox for the Production of Monoterpene-Based Aroma Compounds

Bioprospection studies may be considered a key approach to isolate new strains that convert substrates not yet explore by the current flavor and fragrances biotechnology. Consequently, the modern biotechnology of monoterpenes must complement the cultivation-dependent screening used as the gold standard with functional mining of metagenomes to access the unlimited potential of the environmental pool of enzymes and monoterpene-catabolic pathways [147]. The first reports of a metagenome-based screening to identify genes coding for monoterpene-catabolizing enzymes targeted the microbiome of phytoparasites, which feed on a terpene-rich diet. The metagenome-based analysis of the microbiome from the pinewood nematode *Bursaphlenchus xylophilus* [148] allowed the identification of genes putatively involved in the catabolism of α-pinene, limonene and geraniol, associated with the genera *Pseudomonas, Achromobacter*, and *Agrobacterium*. Similarly, the microbiome of bark beetles has also been reported to comprise monoterpene-metabolizing bacteria [149], from which five detected genes showed homology to α-pinene catabolic enzymes (aldehyde dehydrogenase, an oxidoreductase, an enoyl-CoA hydratase and two hydratases/epimerases) and were associated with the genera *Pseudomonas, Rahnella, Serratia and Stenotrophomonas* [149]. In 2014, the meta-transcriptome of the resin-tolerant microbial community associated with pine tree galls, formed by the moth *Retinia resinella*, detected expression of genes associated with α-pinene catabolism and the *atu* pathway [150], widespread in more than 40 bacterial genera, from which *Pseudomonas* was the most abundant genus.

Although functional characterization is required to assess the catalytic novelty of these genes, which are identified based on homology search against limited databases of monoterpene-catabolizing enzymes, the metagenomics studies are extending the taxonomic knowledge regarding potential biocatalysts.

### 3.2. Holistic Approaches Are the Framework for Monoterpene Biocatalysis À La Carte

The current research has created a catalogue of relevant genes, proteins, pathways and microorganisms, which hold great potential for monoterpene biotechnology and have broadened the biotransformation prospects for virtually unlimited backbone modifications. Nevertheless, despite the efforts to describe the biotransformation of monoterpenes and monoterpenoids in different strains, the majority of the studies lack a holistic characterization of the catabolic pathways and of the monoterpene-catabolizing microorganisms towards proper biotechnological exploitation.

Functional approaches based on high-throughput methodologies (e.g., transcriptomics, proteomics, metabolomics, phenomics and epigenomics) are thus required to characterize and quantify the dynamics underlying the biological processes and molecular interactions of the biological components involved in the biotransformation of a monoterpene substrate. Particularly, the comprehensive understanding of a studied microorganism as a potential cell factory depends on an integrated genotype-to-phenotype overview describing the biological system in order to enable the knowledge base to develop computational and mathematical models, which are essential to predict the dynamic cell behavior and to set the framework for the rational design of metabolic engineering strategies (reviewed in [151] and references therein).

Moreover, a bacterial strain with potential to be used as a cell factory must be easily cultivated under laboratory batch conditions and at a large production scale; grow quickly with minimal nutritional requirements, especially when using low-cost feedstocks to reduce production costs (e.g., monoterpene-rich industrial by-products); and present some degree of solvent tolerance to cope with the hydrophobicity of the monoterpene backbone (referred in Section 2.1 with more detail). The candidate strains must also be prone to genetic manipulation. Although *E. coli* strains are the common workhorse for recombinant approaches due to the well-established knowledge regarding their genetics, metabolism and physiology, the advances of high-throughput approaches and synthetic biology-guided genetic engineering tools have established strains of the genera *Bacillus*, *Pseudomonas*, *Rhodococcus* and *Corynebacterium*, among others, as model organisms [151]. In particular, industrially established strains for the production of biobased chemicals, certified as host–vector systems (e.g., *P. putida* KT2440), may be regarded as promising hosts for the creation of specialized aroma biocatalysts with heterologous functional models [152].

The current metabolic engineering toolbox includes several platforms which allow virtually any kind of genetic manipulation. Genome-wide engineering approaches usually aim at the creation of strains with increased genome stability and reduced metabolic burden by deletion of nonessential genes and mobile genomic loci. The approaches for targeted genetic engineering may include the utilization of standardized vectors for recombinant approaches and controlled protein expression (e.g., pSEVA architecture), biosensor modules for live monitoring of the cell physiology and biotransformation performance, as well as the targeted genetic manipulation or modulation of gene expression (e.g., CRISPR/Cas9-based approaches) [151].

*C. defragrans* DSM 12143, *N. aromaticivorans* DSM 12444, *P. aeruginosa* PA01, *P. citronellolis* DSM 50332, *Pseudomonas fragi* NBRC 3458, *Pseudomonas protegens* Pf-5, *P. putida* F1, *Pseudomonas* sp. M1, *Sphingobium yanoikuyae* B2 and *T. linaloolentis* DSM 12138 are the monoterpene-catabolizing strains whose genome was sequenced and is available for further molecular and functional studies. In particular, the β-myrcene catabolic pathway of *Pseudomonas* sp. M1 is one of the most well characterized cases in the literature. Preliminary work carried out by Soares-Castro et al. [30,144] showed that the different genotypes of the mutant strains M1-N2, M1-C19 and M1-C38 resulted in different metabolic profiles of β-myrcene derivatives, including the differential production of the alcohols ipsdienol, isomyrcenol and *p*-mentha-1,5-dien-8-ol, reported for their odorant and attractant/repellent properties towards several insect species, which could be explored as potential biocontrol agents [153,154]. Therefore, further characterization of these strains may set the adequate knowledge ground to tune the functional modules of the 28-kb genomic island as versatile catalytic tools. Furthermore, a holistic approach would greatly contribute to the development of biotechnological tools towards the bulk production of (i) α-pinene derivatives, such as isonovalal, with *P. rhodesiae* CIP 107491 enzymatic machinery; (ii) a broad array of limonene derivatives using *R. erythropolis* DCL14 enzymes and (iii) α-terpineol by *Sphingobium* sp. NCIMB 11671 enzymatic machinery.

The resilience and biotransformation performance of bacterial cells may be enhanced by complementing the bioprocess with “in situ product removal” (ISPR) approaches (reviewed in [155] and references therein), which allow one to enhance the volumetric productivity and production yield, maintain the chemical stability of monoterpenes and monoterpenoids (e.g., epoxides may be unstable in aqueous phases), reduce biocatalyst inhibition mediated by the accumulation of the product, and decrease downstream processing costs and the amount of waste water generated in the process. One of the most common strategies consists of performing the biotransformation in a two-phase liquid system with an organic top layer of a biocompatible solvent (e.g., *n*-dodecane, *n*-hexadecane, 1-octanol and 1-decanol), which acts as a reservoir for monoterpenes and monoterpenoids that can be recovered by distillation-based methodologies [155]. Two-phase liquid systems have been applied in several studies of monoterpene biotransformation (as exemplified in several case reports in Section 2.3), mainly in fed-batch bioreactor settings, in which higher yields of production were usually obtained, also allowing the utilization of higher amounts of substrate. Furthermore, the in situ removal of more hydrophilic products may be achieved by recirculating the aqueous phase through an adsorbent matrix (e.g., affinity chromatographic resin) [92,156]. Other complementary approaches for optimal ISPR must include online monitoring of chemometric parameters and the mathematical modelling of the downstream product recovery towards the development of a knowledge-base-controlled process.

### 3.3. Coupling the Bacterial Synthesis of Monoterpene Precursors with the Oxidative Biocatalysis into Aroma Compounds

As an alternative strategy, several studies have been focused on the de novo biosynthesis of monoterpene precursors via anabolic pathways [14,157,158]. Monoterpenes can be synthesized de novo from isopentenyl pyrophosphate and dimethylallyl pyrophosphate, obtained by mevalonate (MVA) or 2-C-methyl-D-erythritol 4-phosphate (MEP) pathways, using glucose or glycerol as precursors [159]. The isoprenoid synthesis is mediated by a geranyl pyrophosphate synthase (GPPS), coupled with specific monoterpene synthases for the customized production of the desired monoterpene backbone (Appendix A). The genes used to assembly the biosynthetic pathways in bacterial hosts are usually cloned from various sources, including yeast and plants, which were reported to have often required a codon optimization step, for efficient translation in the host organism.

The biosynthetic strategy with recombinant *E. coli* hosts has been applied for the production of several bicyclic, cyclic and acyclic monoterpenes (Appendix A), such as α-pinene (up to 0.97 g L^−1^ [160]), β-pinene (up to 20 mg L^−1^ [161]), *cis*-sabinene (reaching a maximum concentration of 2.7 g L^−1^ [162]), limonene (up to 2.7 g L^−1^ in a diisononylphthalate two-phase fed-batch setup [163]), β-myrcene (58 mg L^−1^ in a dodecane two-phase system [164]) and geraniol (0.18 g L^−1^ in a decane two-phase system [165]).

Due to the well-studied tolerance of *Pseudomonas* spp. to hydrophobic compounds, Mi and co-workers [166] produced geranic acid in a recombinant *P. putida* DSM 12264 by expression the MVA pathway and a geraniol synthase from *Ocimum basilicum* L. The geraniol produced by the heterologous expression of the geraniol synthase was converted to geranic acid by the enzymatic repertoire of the host. The utilization of glycerol as a precursor led to the production of 0.19 g L^−1^ of geranic acid under fed-batch conditions (Appendix A).

Recombinant cyanobacteria (e.g., *Synechococcus* spp.) have also been studied as hosts for the biosynthetic strategy (Appendix A). These biological systems utilize pyruvate and glyceraldehyde-3-phosphate produced by photosynthesis as precursors for the native MEP pathway [167]. To date, published works on monoterpene biosynthesis by recombinant cyanobacteria reported the production of low limonene titers of up to 4 mg L^−1^ [167].

In a biotransformation-driven framework, the bacterial biosynthesis of monoterpene precursors in recombinant hosts may be promising if coupled to the expression of monoterpene-transforming enzymes, either in the same host, in a mixed culture of strains expressing different enzymatic modules or even in a synthetic biochemistry system. To the best of our knowledge, very few studies have reported such a combined approach. Alonso-Gutierrez et al. [168] coupled the synthesis of limonene via the MVA pathway and activity of a limonene synthase from *Abies grandis* Lindl. in a recombinant *E. coli* DH1, with the heterologous expression of the cytochrome CYP153A6 system (the P450 oxygenase, the ferredoxin and the ferredoxin reductase subunits) of *Mycobacterium* sp. HXN-1500. After several process optimizations (gene codon optimization, CYP153A6 induction dynamics and downstream perillyl alcohol recovery) to improve enzymatic availability and activity, up to 0.44 g L^−1^ of L-limonene was endogenously produced from glucose, and the perillyl alcohol titers, generated from the P450-mediated hydroxylation of L-limonene, reached 0.11 g L^−1^.

Another promising approach relies on the utilization of a cell-free synthetic biochemistry platform by mixing crude preparations or purified enzymes together into self-sustainable metabolic pathways in vitro (Appendix A). By building a biochemistry platform harboring Embden–Meyerhof–Parnas glycolytic and mevalonate enzymes, a system capable of recycling redox equivalents, as well as the α-pinene synthase of *Picea sitchensis* (Bong.) Carr.) or limonene synthase of *Mentha spicata* L. (*wt* or the mutant N345A), Korman et al. [158] were able to obtain higher titers for the production of pinenes isomers (14.9 g L^−1^), limonene (12.5 g L^−1^) and sabinene (15.9 g L^−1^), respectively, with >90% yield using 500 mM of glucose as a precursor. A similar rationale could thus be applied for the tailored oxyfunctionalization of these monoterpenes into value-added derivatives.

## 4. Outlook and Final Considerations

Bacteria exhibit a wide spectrum of evolutionary, functional and metabolic diversity that vastly exceeds that of all other organisms. This metabolic versatility has been exploited for the production of industrially relevant value-added compounds, including top-selling aromas such as vanillin and carvone. The scientific and technological maturation of systems biology approaches and genetic engineering tools has fostered advances in modern microbial biotechnology. As a result, in the last decade, increasing effort has been invested in the development of bioprocesses for the production of aroma compounds.

Nevertheless, the repertoire of identified gene products and their molecular and biochemical characterization is still scarce, especially regarding acyclic monoterpenes such as β-myrcene, which limits the setup of biological systems and these plant volatiles as source of industrially important aromas and novel biomolecules. The development of engineered cell factories regarding monoterpene biotechnology is in its infancy, for which there are no reported studies. As the molecular background associated to monoterpene catabolic pathways is unveiled using holistic characterization approaches, several microbial strains and their versatile functional repertoire are emerging as prospective biotechnological tools, such as *Pseudomonas* sp. M1 and *C. defragrans* DSM 12143, as well as several engineered variants of class I P450 cytochromes (Figure 7).

The current metabolic engineering rationale results from merging systems biology approaches with genome-wide engineering platforms (synthetic biology) and directed evolution strategies, aiming at the tuning and customization of tractable strains to improve biotransformation performance. Moreover, the comprehensive characterization of the full biotechnological potential of the reported biological systems is not only limited by the functional approaches used but also by the maturity of protein and metabolite databases. Currently, the MetaCyc and KEGG databases of metabolic pathways include molecular information of only few monoterpene catabolic pathways, reported in *C. defragrans* (limonene, β-myrcene and linalool), *Rhodococcus* spp. (DL-limonene, DL-carveol, DL-dihydrocarveol and 1,8-cineole), *G. stearothermophilus* BR388 (D-limonene) and *Pseudomonas* spp. (camphor, D-limonene, *p*-cymene and β-citronellol and geraniol).

Therefore, to bridge the knowledge-to-application gap, the research on monoterpene biocatalysis must be driven by and include the following key approaches: (i) the detailed sequencing of the genome of monoterpene-biotransforming strains, followed by accurate curation of gene annotations; (ii) functional characterization of the monoterpene-induced stimulon at the transcriptional, translational, metabolic and physiological level towards the comprehensive understanding of the biological systems and catalytic steps; (iii) the establishment of tools for precise genetic and protein manipulation, including genome-wide engineering (e.g., genome streamlining) and targeting specific functional modules (e.g., catabolic genes, promoter sequences, regulatory elements); (iv) genome-wide in silico metabolic reconstruction based on experimental data, which will allow further refinement of the predictive models; (v) the characterization of the biological properties of new monoterpene derivatives, arising from studies with uncharacterized strains and novel enzymes, which may generate unprecedented scents and flavors; and (vi) updating of the curated data into publicly available databases to reach the whole scientific community. Additionally, future efforts in whole-cell bioprocesses may not only have to focus on designing orthogonal catalytic steps towards metabolic robustness and proper distribution of the carbon flux and redox power of the cell, but also aim at controlling the stability of the desired genetic–physiological trait by limiting phenotypic heterogeneity and cell evolution within the bacterial population.

Furthermore, the establishment of bioaroma production by using monoterpene precursors is mostly in initial stages of development and implementation, also due to the intricate physicochemical properties of monoterpenes and monoterpenoids (e.g., high hydrophobicity, volatility and structural instability), which may constrain the performance of biocatalyst, either as whole cells or enzymes, and the production yield in conventional bioprocess setups based on aqueous catalysis. The successful setup of such biotechnological processes may therefore be dependent on key engineering parameters defined at the genetic (pathway engineering and gene expression), physiological (cytotoxicity, competing pathways and by-product generation) and technological levels (low-cost substrates, volatility, ISPR approaches and downstream processing for product recovery, biocatalyst recycling and reduction of wastes), for which only a systematic approach will allow the identification of the parameters that play a key role in monoterpene biotransformation, and subsequently the development of strategies for bioprocess optimization and sustainability.

## Figures and Tables

**Figure 1 molecules-26-00091-f001:**
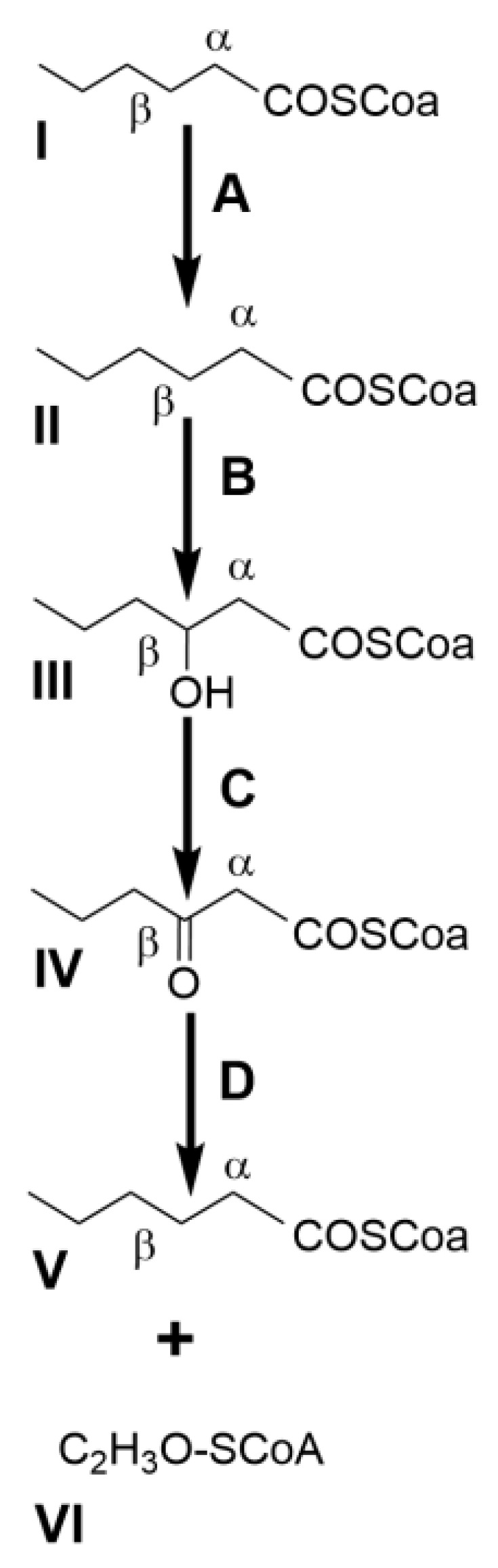
β-Oxidation pathway for the catabolism of coenzyme A moiety (CoA)-activated hydrocarbons in bacteria. After activation of the carboxyl group of the acyl chain (I), an acyl-CoA dehydrogenase catalyses the reduction of the bond between the α,β carbon atoms (A) to form an unsaturated derivative (II), which is further hydroxylated at the β-carbon by a enoyl-CoA hydratase (B). The β-hydroxyl-derivative (III) is oxidized into a β-keto-derivative (IV) by a β-hydroxyacyl-CoA dehydrogenase (C) for subsequent removal of a C2 moiety as acetyl-CoA (VI) by an acyl-CoA acetyltransferase/thiolase (D). The resulting acyl chain is thus shortened by two carbon atoms (V) and remains activated with a CoA moiety for another β-oxidation cycle.

**Figure 2 molecules-26-00091-f002:**
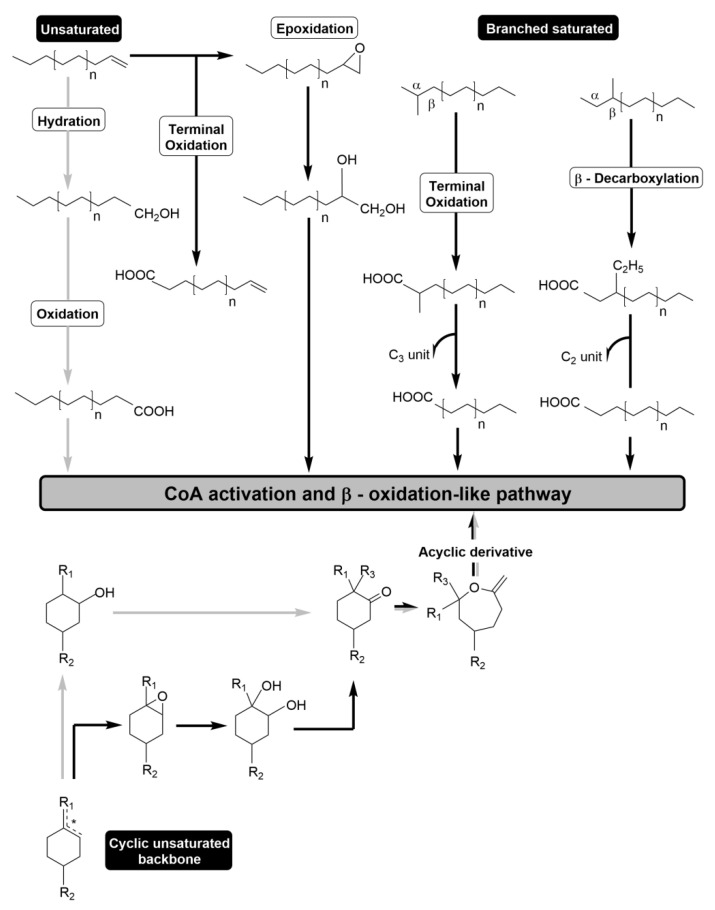
Schematic overview of the catabolic pathways described for acyclic or cyclic medium-chain unsaturated and branched hydrocarbon backbones in bacteria. The catabolism of the unsaturated cyclic backbone is similar to the catabolism of saturated cyclic alkanes, once the molecule is oxidized into a ketone. After being converted to a *n*-alkane-like backbone, the hydrocarbon intermediates are channeled to the central metabolism by β-oxidation-like pathways. Grey arrows depict mechanisms described for anaerobic bacteria, and black arrows depict mechanisms described for aerobic bacteria. R1, R2 and R3 may represent hydrogen atoms or any hydrocarbon-related group.

**Figure 3 molecules-26-00091-f003:**
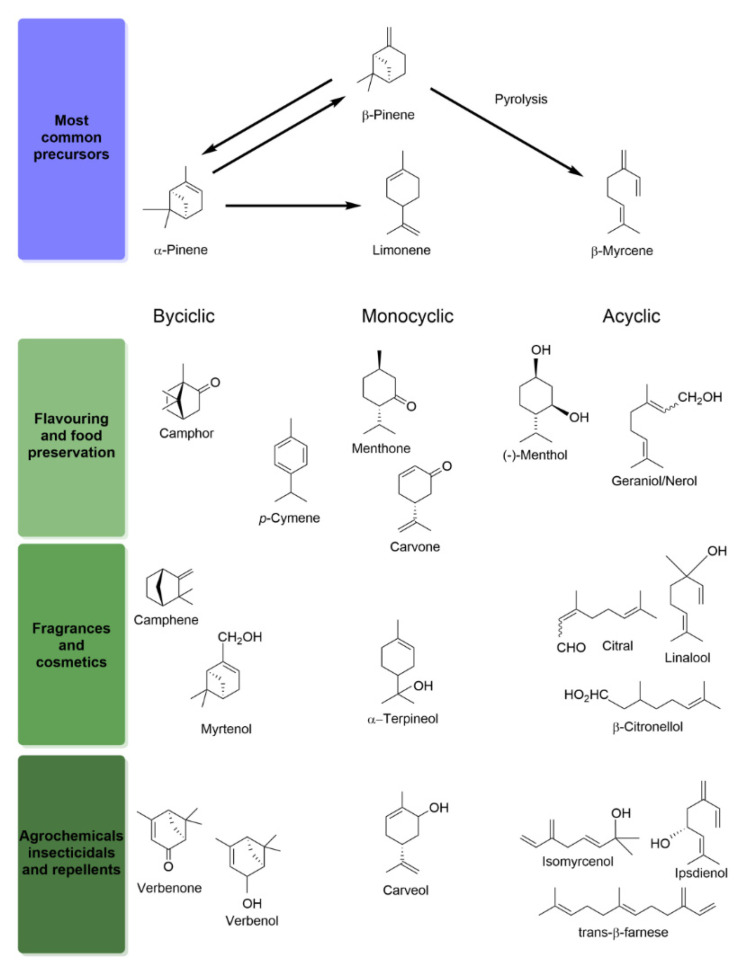
Common derivatives obtained by the catalysis of pinenes isomers, limonene and β-myrcene, with applications in the food and flavoring industries, as fragrances and in the production of agrochemicals.

**Figure 4 molecules-26-00091-f004:**
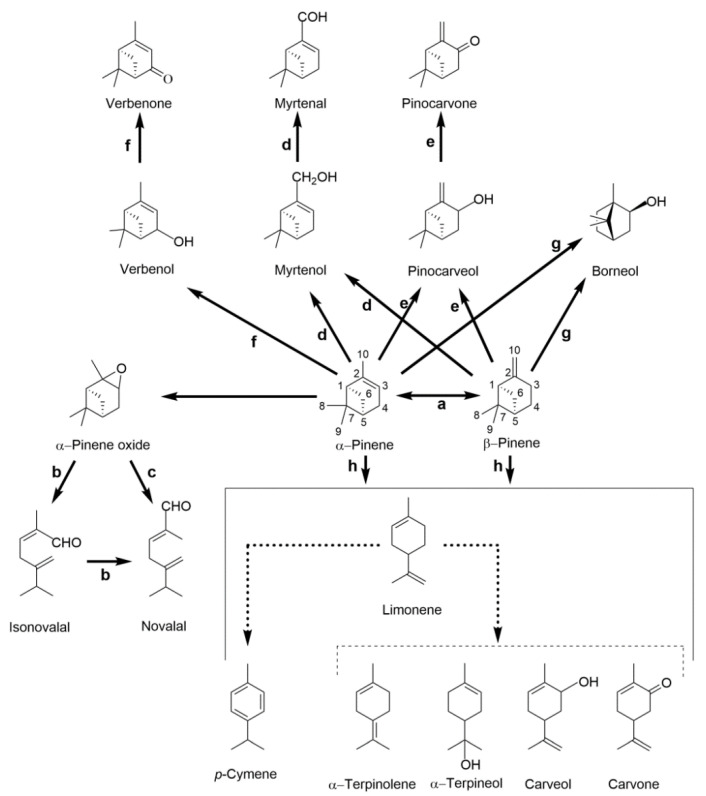
Most common catabolic pathways of α- and β-pinene described in bacteria. Lettered pathways correspond to the following literature: isomerization (**a**) reported by [70] (**b**,**c**) reported by [71,72,73,74,75]; (**d**,**e**) reported by [70,76,77,78]; (**f**) reported by [79]; (**g**,**h**) reported by [30,70,73,76,77,78,79]. Dashed arrows show putative metabolic reactions based on the metabolic profiles described in the literature.

**Figure 5 molecules-26-00091-f005:**
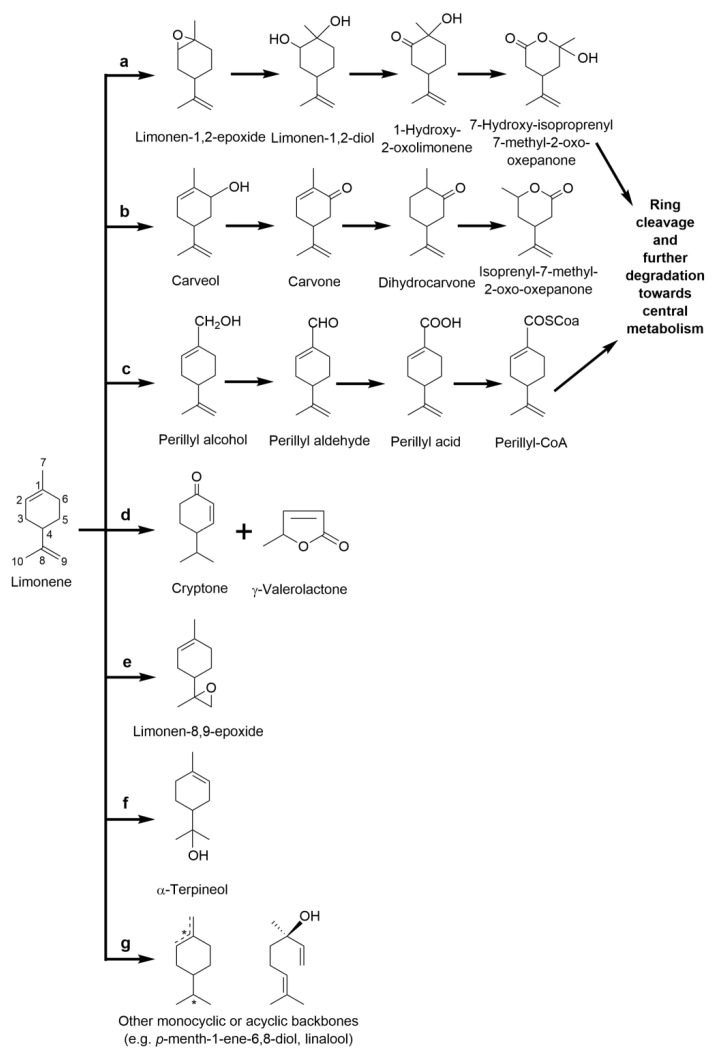
Most common catabolic pathways of limonene described in bacteria. Lettered pathways corresponded to the following literature: (**a**) reported by [30,86]; (**b**) reported by [70,86,88]; (**c**) reported by [38,89,90,91,92,93]; (**d**) reported by [91]; (**e**) reported by [94]; (**f**) reported by [70,91,95]; (**g**) reported by [93,96].

**Figure 6 molecules-26-00091-f006:**
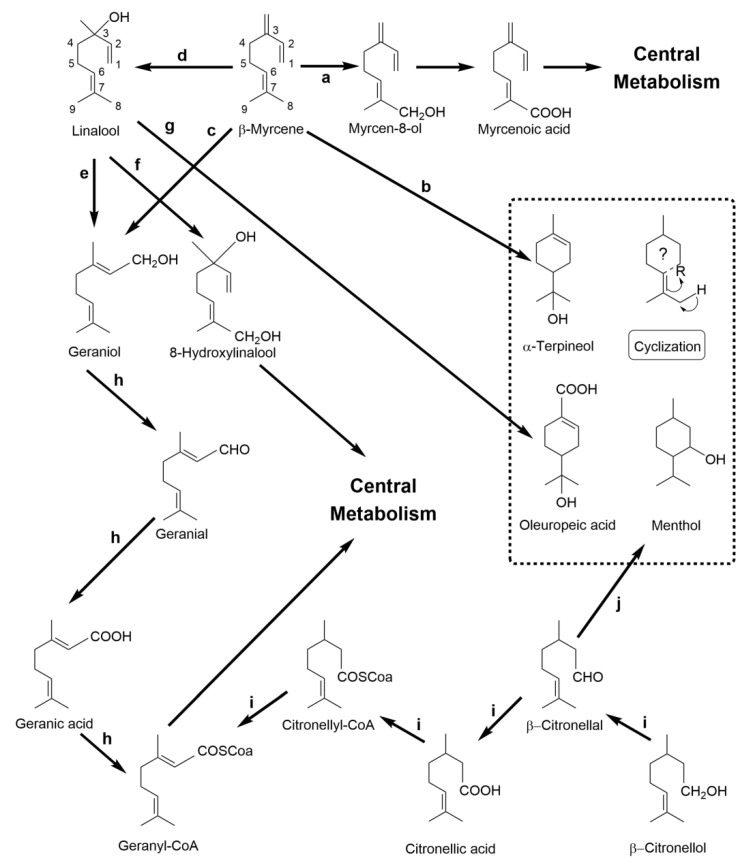
Catabolic pathways of the acyclic monoterpenes described in bacteria. Lettered pathways corresponded to the following literature: (**a**) reported by [30,109]; (**b**) reported by [111]; (**c**) reported by [112]; (**d**) reported by [58,111]; (**e**) reported by [30,113]; (**f**) reported by [30,114,115,116]; (**g**) reported by [117]; (**h**) and (**i**) reported by [30,65,116,118]; (**j**) reported by [119].

**Figure 7 molecules-26-00091-f007:**
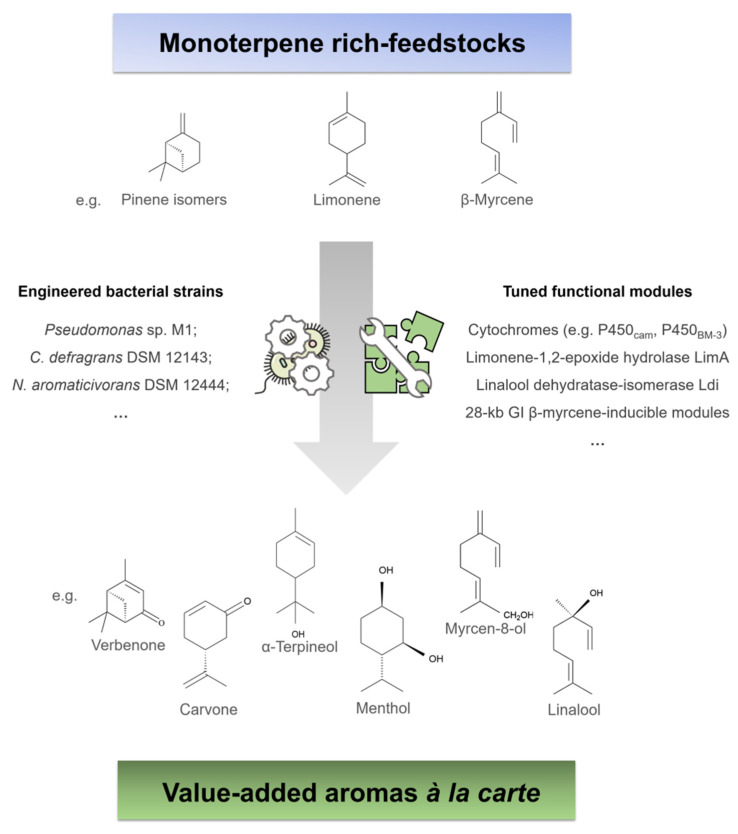
Schematic overview of the current strategies envisaging the bioproduction of aroma compounds à la carte. The systems biology approaches are increasing the molecular background underlying monoterpene catabolic pathways; thus, microbial strains which have been characterized by functional genomics methodologies (e.g., *Pseudomonas* sp. M1, *Castellaniella defragrans* DSM 12143), as well as several monoterpene-transforming functional modules, are emerging as prospective biotechnological tools for the valorization of monoterpene-rich feedstocks.

**Table 1 molecules-26-00091-t001:** Examples of the most relevant compounds currently used by the flavor and fragrance industries (based on the information retrieved from the website of the companies, as well as from references [5,12,13].

Compound	Aroma	Observations of Biotech Production
*Phenolic aldehydes*		
vanillin	vanilla aroma	biotech production established by, e.g., Evolva-IFF, Solvay, Mane, Shangai Apple, BASF, Isobionics
safranal	saffron aroma, sweet, spicy, floral odor with a bitter taste	biotech production announced by, e.g., Evolva
*Lactones*		
γ-decalactone	fruity, peach-like aroma	biotech production established by, e.g., BASF, Symrise
γ-undecalactone	fruity, sweat peach-like aroma	-
*Ketones*		
2-heptanone	fruity, cinnamon, banana-like	-
α- and β-Ionone	woody, raspberry-type, floral, violet-like odor	-
nootkatone	citrusy notes and grapefruit-like aroma	biotech production established by, e.g., Allylix (Evolva), Isobionics, Oxford Biotrans
*Alcohols*		
1-octen-3-ol	sweet, earthy, herbaceous floral notes, reminiscent of lavender	-
*Carboxylic acids*		
citric acid	acid taste; odorless	-
*Esters*		
ethyl butanoate	sweet and pineapple-like aroma	-
*Essential Oils*		
orange peel oil	orange aroma	-
lemon peel oil	lemon aroma	-
eucalyptus oil	camphoraceous odor, spicy, cooling taste	-
peppermint oil	odor of peppermint, cooling, minty, menthol, sweet taste	-
spearmint oil	minty, carvone-like, cooling, candy, spicy	-
*Monoterpenes*		
α-pinene	terpy, citrus and spicy, woody pine and turpentine-like with a slight cooling camphoraceous nutmeg-like note	-
β-pinene	cooling, woody, piney and turpentine-like with a fresh minty, eucalyptus and camphoraceous note	-
1,8-cineole	cooling, fresh, oily, green, spicy, pine-like	-
limonene	(+)-limonene has an orange-like odor (−)-limonene has a more harsh turpentine-like odor with a lemon note	-
(−)-menthol	minty, coolant odor	-
menthone	minty, cooling, sweet, peppermint, camphoraceous aroma with a green herbal anise nuance	-
carvone	(*R*)-(−)-carvone has a spearmint aroma (*S*)-(+)-carvone has a caraway aroma	-
α-terpineol	pine odor, floral aroma	-
β-myrcene	terpy, herbaceous, woody odor with a mango-like nuance.	-
linalool	floral, fresh, sweet, citrus-like aroma	-
citronellol	rose-like scent	-
citral	lemon, peely, citrus, floral with woody and candy notes.	-
geraniol	rose-like, sweet, fruity aroma	-
*Sesquiterpenes*		
α-farnesene	dry woody, green leafy, herbal and floral nuance	biotech production established by, e.g., Amyris-Antibióticos S.A
(+)-valencene	sweet, fresh, grapefruit-like aroma	biotech production established by, e.g., Allylix (Evolva), Isobionics

**Table 2 molecules-26-00091-t002:** Biocatalysts currently used in industrial applications, derived from research of monoterpene catabolism in bacteria.

Whole-Cell Biocatalyst	Substrate	Product	Ref.
*Pseudomonas* sp. TK2102	Eugenol	vanillin	[42]
(JP patent 5227980)			
*Pseudomonas putida* ATCC55180	Eugenol	vanillin	[43]
(US patent 5128253)	Ferulic acid		
*Pseudomonas* sp. NCIB 11671	α- and β-Pinene	(−)-carvone (spearmint aroma)	[44]
(US patent 4495284)			
**Enzymatic Biocatalyst**	**Substrate**	**Product**	**Ref.**
Commercial lipase AK from *Pseudomonas fluorescens* (Amano Enzyme Inc.)	(±)-menthol,	(−)-menthyl acetate for the production of (−)-menthol	[45]
	(±)-neomenthol,		
	(±)-neoisomenthol,		
	(±)-isomenthol		
Commercial lipase PS from *Burkholderia cepacia* (Amano Enzyme Inc.)	(±)-isopulegol isomers and vinyl acetate	(−)-isopulegol acetate for the production of (−)-isopulegol	[46]
Commercial lipase PS from *Burkholderia cepacia* (Amano Enzyme Inc.)	(±)-mentholand vinyl acetate	(+)-menthol and(−)-menthyl acetate for the production of (−)-menthol	[46]
Commercial esterase from *Bacillus subtilis* ECU0554	(±)-menthol esters	(−)-menthol	[47]

## Data Availability

No new data were created or analyzed in this study. Data sharing is not applicable to this article.

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
