# Peer review of "Current Advances in the Bacterial Toolbox for the Biotechnological Production of Monoterpene-Based Aroma Compounds"

_molecules, 2020, doi:10.3390/molecules26010091_

Round 1

Reviewer 1 Report

The review is in line with the objectives have been proposed

The authors have linked microbial biotechnology over chemical methods with the demand for monoterpenes by the industries.

The advantages of microbial biotechnology over chemical methods highlighted in the article have been evidenced throughout the text.

The illustrations are simple and allow the reader to focus on the content of the text.

I agree with the publication of the Review: “Current advances on the bacterial toolbox for the biotechnological production of monoterpene-based aroma compounds”

Author Response

The authors thank the reviewer1 for the comments. In the revised version of the manuscript, typos and spelling were checked and corrected.

Reviewer 2 Report

This manuscript written by P. Soares-Castro, F. Soares and P. M. Santos describes the utilize of bacterial cells and bacterial enzymes in the biotechnological production of monoterpene-based aroma compounds. The reviewer believes that this manuscript is well written and interesting and is suitable for publication in Molecules. However, the reviewer recommends publication after some revision as follows:

1). The names of the plants and the microorganisms, the symbols "trans/cis p-" etc. configurations (R) and (S) should be written in italic, e.g.

L143: R-(+)-limonene should be corrected to (R)–(+)-limonene, L372, 373: trans-carveol. trans-verbenol, trans-sobrerol, L377: R-(+)- and S-(-)-limonene, L384: p-cymene, L403: degrade (R)- and (S)-limonene, L405: generates (R)- and (S)-carveol. DCL14 cells can oxidize both (R)- and (S)-stereoisomers, L412: 6-hydroxy-5isopropenyl-2-methylhexanoate from (1R,4R)- or (1S,4R)-(iso-)dihydrocarvone. In the name of compound put a hyphen between 5-iso…, L444-445: of R-(+)-α-terpineol, from R-(+)-limonene. L520: (+)-limonene, L.530: to (S)-(+)-linalool and isomerization of (S)-(+)-linalool to geraniol, L690: p-mentha-1,5-dien-8-ol,

L681-686: the names of the bacteries should be written in italic.

2) The names of the plants should be written correctly: the names of the authors are missing, e.g. L89: Ricinus communis, L736: Ocimum basilicum. L750: Abies grandis, L761-762 Picea sitchensis and Mentha spicata, etc.

3) The bacterial strains should be written correctly - full name, e.g. L448: Pseudomonas sp. L540: Pseudomonas convexa (j in Figure?) , Figure number lost. Please check the entire manuscript!

4). The names of bacteria or enzymes used for the first time should have a full name, e.g. Bacillus pallidus , later abbreviations may be used, e.g. B. pallidus. Please check the entire manuscript!

Author Response

The authors thank the reviewer2 for the comments. In the revised version of the manuscript, typos and spelling were checked and corrected. All the suggestions and comments of reviewer2 were considered in the revised version of the manuscript.

In particular, regarding the comments made, please see the attachment.

Reviewer 3 Report

This is an interesting review gathering different works on biotechnological production of monoterpene-based aroma compounds. In the literature there are numerous papers dealing with the biological production of specific monoterpenes.

Given that this review gives detailed general information on the literature about biological production of aromatic monoterpenes and contains an extend introduction on the field, in my opinion, it could be useful and be accepted for publication on Molecules

Author Response

The authors thank the reviewer3 for the comments. In the revised version of the manuscript, typos and spelling were checked and corrected.